# Clinical trial and detection of SARS-CoV-2 by a commercial breath analysis test based on Terahertz technology

Meila Bastos De Almeida[1], Regina Aharonov-Nadborny[2], Eran Gabbai[2], Ana Paula Palka[1], Leticia Schiavo[1], Elis Esmanhoto[1], Irina Riediger[3], Jaime Rocha[4], Ariel Margulis[2], Marcelo Loureiro[1], Christina Pettan-Brewer[5], Louise Bach Kmetiuk[6], Ivan Roque De Barros-Filho[6], Alexander Welker Biondo[6]*

1 Paraná Institute of Technology—TECPAR, Curitiba, Paraná State, Brazil, 2 TeraGroup Terahertz Ltd, Herzliya, Tel Aviv District, Israel, 3 Paraná State Reference Laboratory, São Jose dos Pinhais, Paraná State, Brazil, 4 Department of Infectious Diseases, Pontifical Catholic University, Curitiba, Paraná State, Brazil, 5 Department of Comparative Medicine, School of Medicine, University of Washington, Seattle, WA, United States of America, 6 Department of Veterinary Medicine, Federal University of Paraná State, Curitiba, Paraná State, Brazil

* abiondo@ufpr.br

**Data Availability Statement:** All relevant data are within the paper and its Supporting Information files.

## Abstract

Public health threats such as the current COVID-19 pandemics have required prompt action by the local, national, and international authorities. Rapid and noninvasive diagnostic methods may provide on-site detection and immediate social isolation, used as tools to rapidly control virus spreading. Accordingly, the aim of the present study was to evaluate a commercial breath analysis test (TERA.Bio®) and deterministic algorithm for detecting the SARS-CoV-2 spectral signature of Volatile Organic Compounds present in exhaled air samples of suspicious persons from southern Brazil. A casuistic total of 70 infected and 500 non-infected patients were sampled, tested, and results later compared to RT-qPCR as gold standard. Overall, the test showed 92.6% sensitivity and 96.0% specificity. No statistical correlation was observed between SARS-CoV-2 positivity and infection by other respiratory diseases. Further studies should focus on infection monitoring among asymptomatic persons. In conclusion, the breath analysis test herein may be used as a fast, on-site, and easy-to-apply screening method for diagnosing COVID-19.

## Introduction

The current COVID-19 pandemic, caused by the respiratory virus SARS-CoV-2, has demonstrated the importance of rapid and focused use of financial and human resources [1]. Healthcare systems have been continuously challenged worldwide for decision-making guidance on social distancing, self-isolation, curfew, lockdown, and other preventive measures against virus spreading and mortality [2, 3]. A reliable, rapid, noninvasive, breathalyzer and on-site test, particularly for detecting asymptomatic people, could be used to avoid such lossmaking measures, thereby allowing uninterrupted working of commercial companies and public services [4–7].

**Funding:** Brazilian National Council for Scientific and Technological Development -CNPq (material support) (2020-1/402341- AWB and CPB). The funders had no role in study design, data collection and analysis, decision to publish, or preparation of the manuscript.

**Competing interests:** The authors have declared that no competing interests exist.

The SARS-CoV-2 spreading has resulted in a persistent pandemic to date, mainly due to fast transmission among people in close contact [8], beginning 2–3 days before the symptom onset [9], and predominantly through respiratory transmission [10]. Although the reverse-transcription polymerase chain reaction (RT-qPCR), mostly applied on nasopharynx swab samples, has been reportedly considered to be the gold-standard diagnostic method for SARS-CoV-2 by the World Health Organization (WHO) [11], false negative rate has widely varied from higher in the first 5 days (up to 67%) to lower on day 8 after exposure (21%) [12]. Asymptomatic SARS-CoV-2 carriers (varying from 5% to 80%) may remain unknown spreaders in their communities and during traveling [13]. Thus, reliable SARS-CoV-2 diagnosis has been crucial for rapid detection, interruption of transmission and isolation [11].

In addition to swab samples, sputum and exhaled air have supposedly been used in medical diagnosis due to air dispersion of volatiles organic compounds (VOC) as a result of the patient's metabolism [14]. Considering the noninvasiveness of exhaled air testing, such sampling may potentially become a very convenient diagnostic method. Analyses on VOCs have become well established for differentiation between healthy and sick human samples, particularly in pulmonary conditions, applied in patients as noninvasive diagnostic and monitoring tool [15].

Besides normally exhaling different VOCs, representing metabolic processes of body physiological biochemistry, human respiration may also contain specific biomarkers as result of internal chemistry changes during systemic disorders [16]. Endogenous gases may be also identified in the VOC analysis of exhaled air, including methane, isoprene, acetone, and aldehyde [15], mostly detected by gas chromatography and infrared spectroscopy [17].

As breath analysis tests (BATs) may provide faster on-site detection rate than nasopharynx samplings [18], gas chromatography-mass spectroscopy [20] and infrared spectroscopy [15, 16] have been established as the two most used detection methods. However, gas chromatography has been an expensive and non-portable method [19], while humidity of exhaled air samples may interfere with infrared spectroscopy results [15].

The emerging technology based on Terahertz (THz) radiation has been able to detect biofingerprints such as VOCs, viruses, bacteria, and inorganic material of exhaled air [20, 21]. Terahertz (THz) waves are located in the electromagnetic radiation between the microwave and the infrared spectrum [17, 22]. THz radiation has consisted of non-ionizing electromagnetic waves considered safe for human subjects and operators [23–26] In addition, studies have shown that the association of algorithms and artificial intelligence may increase the diagnostic effectiveness [27].

The growth of THz technology has been directed towards rapid notification of tests by on-site operators, and screening between SARS-CoV-2 negative and positive individuals [28]. Use of metasensors may provide more rapid and precise screening and detection of key components of viral entry such as the receptor-binding domain (RBD) of the spike (S) protein in symptomatic or asymptomatic carriers [27, 29]. In such scenario, THz approaches may lead to advances on a fast, accessible, and highly sensitive diagnostic tests for SARS-CoV-2 and other pathogenic viruses [17].

A new commercially available breath analysis test (TERA.Bio®, TeraGroup Terahertz Ltd, Herzliya, Israel) may also be able to identify the presence of specific VOCs in exhaled air samples from infected individuals with SARS-CoV-2, when used in association with a special proprietary algorithm system. Accordingly, the aim of the present study was to evaluate a commercial expired air analysis test (TERA.Bio®) based on THz technology for VOC identification in exhaled air samples from SARS-CoV-2 infected and non-infected individuals.

## Material and methods

### Study sample size and data validation

A validation study on the performance of TERA.Bio® for making SARS-CoV-2 diagnoses was conducted using clinical samples (Fig 1). The validation was designed as a prospective uncontrolled cross-sectional study, with a single arm, to analyze this commercial BAT for diagnosing SARS-CoV-2. RT-qPCR was used as the gold-standard method for evaluating test sensitivity and specificity. The RT-qPCR results used in this work were obtained at the Paraná State Reference Laboratory (Lacen), during routine analysis. This was a study, where Lacen researchers had no access to data generated by TeraGroup researchers, and TeraGroup researchers had no access to data generated by Lacen researchers. Only researchers from the Paraná Institute of Technology were able to compile the data and share the results after statistical analyses.

This study, as it required direct contact with patients, had to be notified and approved by the National Research Ethics Commission of Brazil (protocol number 35555720.7.0000.5225), The Brazilian Registry of Clinical Trials (protocol number U1111-1257-4565) and the National Health Surveillance Agency (protocol number 25352.109400/2020-72) (S1 File).

The present study was conducted in Curitiba (25°25'47" S, 49°16'19" W), capital of Parana State and the ninth most populated metropolitan area in Brazil, with an estimated population of 3,693,817 habitants. Symptomatic and asymptomatic individuals from Curitiba and metropolitan area were simple randomly selected in a convenience sampling (attendance routine) and evaluated at the referral unit of Oswaldo Cruz Hospital between September 9 and September 22, 2020. The inclusion criteria were that these individuals should be outpatient cases, older than 18 years of age, who had signed a free and informed consent statement; and that sampling for both the RT-qPCR test and the SARS-CoV-2 BAT were performed within the same day. Patients were excluded from the present study when younger than 18 years old (ethics issue and considering their lower exposure to SARS-CoV-2 infection and COVID-19 manifestation), hospitalized inpatients, patients whose BATs for SARS-CoV-2 were analyzed more than six hours after sampling, and patients who did not signed the consent statement for any reason.

Although measurements in the THz spectrum were affected by environmental conditions such as temperature and humidity, such problem was solved by maintaining constant environmental conditions when operating the device (air-conditioned room at a constant temperature of 22–23˚C and 30–40% humidity). In addition, sampling room air was measured in separate tests as part of system performance evaluation, presenting insignificant variability.

The assumed positivity of RT-qPCR for sampling calculation in the clinical trial herein was 25%, based on epidemiological COVID-19 reports of Curitiba city, which had an estimated population of approximately 1.9 million habitants at the time. Aiming at a type I error (α) of 0.05, and an estimated precision of 0.05, a minimum sample size of 400 subjects was obtained to statistically evaluate the BAT test, considering a 90% sensitivity as minimum for a useful diagnostic test [30]. Representing a consecutive, casuistic, random, and homogeneous sampling, collected from all subjects who presented themselves at the hospital in the period, a total of 570 successful samplings were included in the present study, with statistical power of 0.819.

Each subject (both SARS-CoV-2 infected and non-infected individuals) was asked to blow five times into a disposable Teratube, which retained the exhaled air sample for subsequent VOC detection. The Teratube and nasopharyngeal swab samples were individually labeled to ensure traceability. The samples were subsequently tested using their spectral characteristics in the THz band, using the "BioStation"®.

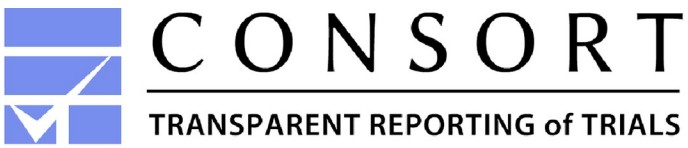

## CONSORT 2010 Flow Diagram

**Enrollment**

Assessed for eligibility (n= 599)

Excluded  (n=29)
♦ Not meeting inclusion criteria (n=  )
♦ Declined to participate (n=  )
♦ Other reasons (n=29)

Randomized (n=570)

**Allocation**

Allocated to intervention (n=570)
♦ Received allocated intervention (n=570)
♦ Did not receive allocated intervention (give reasons) (n=  )

Allocated to intervention (n=570)
♦ Received allocated intervention (n= 570)
♦ Did not receive allocated intervention (give reasons) (n=  )

**Follow-Up**

Lost to follow-up (give reasons) (n=  )

Discontinued intervention (give reasons) (n=  )

Lost to follow-up (give reasons) (n=  )

Discontinued intervention (give reasons) (n=  )

**Analysis**

Analysed  (n=570)
♦ Excluded from analysis (give reasons) (n=  )

Analysed (n= 570)
♦ Excluded from analysis (give reasons) (n= 0)

**Fig 1. Flow diagram for the BAT SARS-CoV-2 test validation.**

## Commercial breath analysis test

The commercial breath analysis test (TERA.Bio®) used herein was composed of a reading equipment associated to a proprietary deterministic algorithm. In short, a portable electronic station, called the "BioStation"®, was used to analyze the biomaterials of respiratory aerosols collected in internal "Teratubes". The BioStation, a device with embedded and integrated technology was controlled by the "Terasystem", a THz technological diagnostic platform built-in software called "BioPass"®, which used a scanner ("TeraScanner") to analyze the spectral signatures of biomaterials using the spectroscopy system (Terasystem).

The main components of the BioStation include two distributed-feedback class IIIB lasers, electric temperature control units, a control unit, a power unit, and photo mixers. The laser beam was used to modulate a photocurrent at a tuned THz frequency, illuminating the photomixer. The THz beam travels through the test sample, the signal is received at the photo-mixer receiver, analyzed, and associated with the algorithm, as previously described by others [17].

The algorithm, based on data collected during in-vitro validation and clinical studies, identifies the spectrum area where biomarkers (biomaterials of respiratory aerosols) were associated with SARS-CoV-2 location. Built on an additional repetition of breath scan features, applied on a machine learning (ML) of a statistical model, the system has become able to differentiate between clear (negative patient) and not clear (patient with suspected SARS-CoV-2 infection) breath samples, from which the algorithm was set. An illustration of ML and algorithm working was presented (Fig 2) to demonstrate separation capability in a specific area, where any result >>0.5 or <<0.5 (further away from 0.5) was considered to represent good differentiation between clear and non-clear samples. The algorithm used by the software for the analysis was adjusted for the target sample during the equipment calibration phase (proprietary property).

The Teratube, an accessory of the BioStation, was used to collect and retain exhaled air samples for further identification of VOCs in exhaled air samples from SARS-CoV-2 infected and non-infected individuals, a THz spectroscopy scanner. The Teratubes are composed of portable and disposable polypropylene-based tubes with a disposable melamine foam-based membrane, used herein for the best absorption of the electromagnetic wave. Membranes were made for single-use and discarded as biohazardous materials after use and analysis.

The study protocol has included non-hospitalized patients who were referred by the Curitiba city health professionals after suspicious RT-qPCR results. Thus, exhaled air samples and

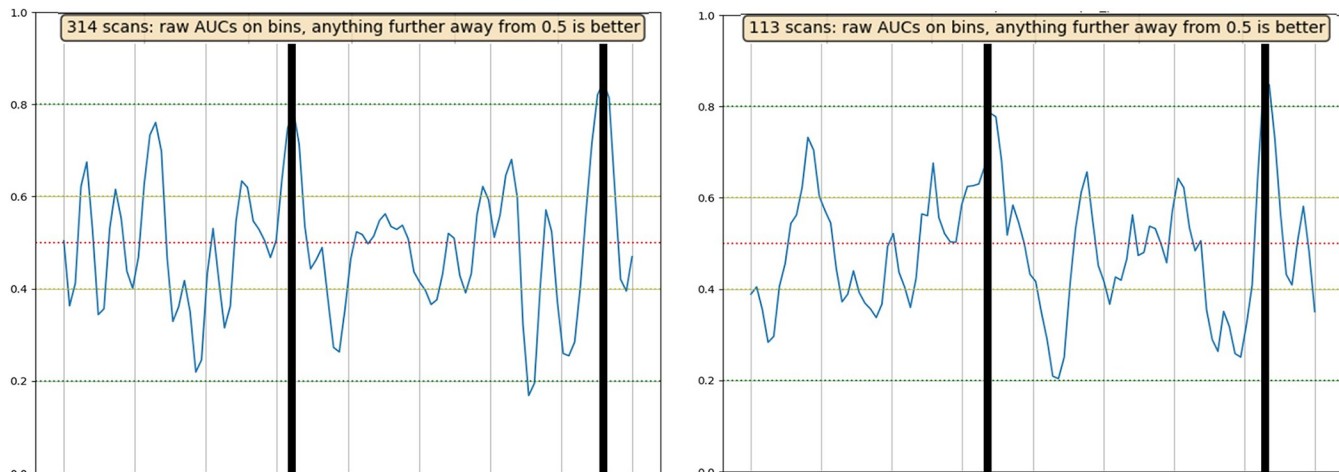

**Fig 2. Separation capabilities between clear and non-clear samples based on biomaterial spectral signatures, where the X-axis was the THz frequency range, and the Y-axis was the measurable index for differentiation ability (0.0 to 1.0).**

nasopharyngeal swab samples were concomitantly collected and tested by the official city service. Each patient sampled by nasopharyngeal swab was also given a dischargeable kit of exhaled air analyzer containing an identified sampling device, which was uncovered in both sides with opened capsule and the melamine membrane internally coupled. Patients were then asked to keep a 2-meter distance from the operator, swallow all saliva in the mouth and deeply blow the device for 5 times. After blown, the patient was asked to close both device sides with the correspondent lids (within the kit) until hear a click noise of sealed lid, capturing the exhaled breath constituents inside the device. The operator received the device, inspected the complete lid closure, and disinfected the external parts with 70% alcohol. After taken to the laboratory, device was opened and capsule with internal membrane removed and inserted into the equipment for analysis. Samples were collected up to 6 hours maximum after sampling, which insured best readings due to membrane integrity, ideal absorption of electromagnetic wave and wave propagation into the exhaled air sample. After analysis, device was discharged as hospital contaminated garbage.

## RT-qPCR test procedure

Nasopharyngeal secretion from patients was obtained by a rayon swab and stored in viral transportation media (Laborclin Laboratory Products, Pinhais, Brazil), placed at 4C, and stored at the Paraná State Reference Laboratory until testing, following standard protocols previously established [31]. Confounding coinfections due to other viruses were also investigated using the same protocols described above and included influenza A, influenza B, human coronavirus 229E, human coronavirus NL63, human coronavirus HKU1, human coronavirus OC43, adenovirus, respiratory syncytial virus, metapneumovirus, rhinovirus, bocavirus, enterovirus, parainfluenza type 1, parainfluenza type 2 and parainfluenza type 3. SARS-CoV-2 detection was carried out using a commercially available molecular kit (BIOMOL One-step/COVID19, Molecular Biology Institute of Paraná [IBMP], Curitiba, Brazil), following the manufacturer's protocol [32].

The RT-qPCR results from these subjects were notified into the Brazilian National Reporting System and immediately delivered to patients by the primary care team. However, results were not available to the TeraGroup team until the end of the analysis, ensuring the reliability of results.

## Statistical analysis

Sociodemographic, epidemiological, and clinical characteristics were presented as percentages, arithmetic means and standard deviations. Differences in proportions were compared by chi-square test with calculated 95% confidence intervals. The results were presented in contingency tables, allowing the calculation of sensitivity, specificity, and predictive values for the BAT. Assessment of association between categorical variables, and BAT and RT-qPCR tests was performed by Fisher's exact test. Mean differences in variables were compared by p-values of Student's unpaired t-test, assuming statistical significance when $p < 0.05$. Precision of the BAT was assessed using ROC curve analysis.

In addition, machine learning was based on "training data" that had been collected prior to the study herein. Over 4,000 subjects were tested using the BATs at several locations worldwide including Brazil. The THz spectrum features for healthy and infected (negative and positive) subjects were processed using ML techniques, to establish a mathematical algorithm for COVID-19 classification. The statistical analyses were performed using the SPSS version 17 software.

### Equipment calibration

As previously mentioned, a first stage (previous study) of ML was carried out which determined the algorithm, and in this stage, it was observed an instability was detected through averaging all the scans from each BioStation and then calibrating the equipment with known clinical samples (301 positive and 296 negative samples in Brazil). After observation that when correlating the air sweep into closed and temperature-monitored chamber within a controlled operating environment, the system checks whether the lasers were reproducing the same frequencies and whether the signal to noise ratio level was kept within specification limits, the instability was eliminated. Additionally, the water signature has been integrated into the THz spectrum of the Terasystem. This constitutes a known parameter for indication of THz. This "high-quality" scanning method that has been integrated into Terasystem (and subsequently into the BioStation) has provided a stable reference point with no shift in the X-axis.

## Results

Although the protocol was initially approved for 400 participants, sample size was increased (after approval by the regulatory institutions) to accommodate a drop of positive cases at the time of survey.

Additionally, 29 patients were excluded due to lack of written agreement or tracking information. In overall, tests were applied to breath samples from 500 unconfirmed and 70 confirmed infected persons.

The clinical, epidemiological, and sociodemographic data of all subjects included in the present study were obtained through a self-report form, which was analyzed and presented, contributing to the ML stage (Table 1, S1 Dataset).

Statistical analysis was based on ML data and resulted in identification of several spectral regions (hot spots) areas. Each of them had a spectral signature for COVID-19 subjects, for example: 500–550 GHz; 900–950 GHz and 1100–1150 GHz as illustrated (Fig 3). In addition to the hot spots, the cohort difference between healthy and infected people was based on the algorithm. Differences were identified by this model of ML techniques and translated into positive or negative results. Although all methods and results have been provided, ML techniques remain a proprietary product (TeraGroup Terahertz Ltd., Herzliya, Tel Aviv District, Israel).

### Analytical performance of BAT for SARS-CoV-2

The results from the RT-qPCR test and SARS-CoV-2 BAT for all subjects was presented (Table 2). Using RT-qPCR method as the gold standard, the commercial BAT method was found to have 92.7% sensitivity (CI 84.1–97.6%) and 96.0% specificity (CI 93.9–97.5%). The positive predictive value (PPV) and negative predictive value (NPV) were determined as 76.5% (CI 66.3–85.0%) and 99.0% (CI 97.6–99.7%), respectively. Fisher's exact test showed a statistically significant association between results ($P < 0.0001$). The area under the ROC curve of the total sampling was 0.94 (SD 0.19; CI 0.91–0.98) (Fig 4).

Considering only the symptomatic patients and using RT-qPCR as the gold standard, BAT presented 92.4% sensitivity (CI 83.2–97.5%) and 95.9% specificity (CI 93.1–97.7%). The PPV was determined as 81.3% (CI 70.7–89.4%) and the NPV as 98.5% (CI 96.5–99.5%). Fisher's exact test showed a statistically significant association between results ($p < 0.0001$). The area under the ROC curve calculated only with symptomatic patients was 0.94 (standard error 0.20; CI 0.90–0.98).

Given that the optimum sampling window of opportunity for RT-qPCR has been reported to comprise the first seven days with symptoms, the data were stratified for analysis including the symptomatic patients who had shown symptoms for up to seven days. In such scenario,

**Table 1. Sociodemographic, epidemiological, and clinical data of the 570 subjects included in the present study.**

| Features* | Confirmed cases (n = 70) | Unconfirmed cases (n = 500) | P-value[a] |
|---|---|---|---|
| Age (years) | 38.6 ±11.1 | 37.1±12.5 | 0.345 |
| Males | 41 (58.0%) | 233 (46.6%) | 0.060 |
| Body mass index (kg/m$^2$) | 27.6 ± 4.7 [d] | 25.9 ± 4.5 | 0.004 |
| Non-white race | 25 (40.3%) [a] | 71 (15.4%) | 0.0001 |
| Travel history | 07 (10%) | 81 (16.7%) | 0.109 |
| Suspicious case contact | 35 (50%) | 218 (43.6%) | 0.232 |
| Confirmed case contact | 32 (45.7%) [a] | 156 (31.2%) | 0.026 |
| Attended at health service | 20 (28.6%) | 185 (38.2%) | 0.090 |
| Symptomatic | 66 (94.3%) [a] | 338 (67.6%) | 0.0001 |
| Days of symptoms [b] | 5.4 ± 2.6 (75.7%) | 5.1 ± 3.4 (36.4%) | 0.468 |
| Fever or chills | 55 (78.6%) [a] | 93 (18.6%) | 0.0001 |
| Myalgia or arthralgia | 63 (90.0%) [a] | 129 (25.8%) | 0.0001 |
| Headache | 44 (62.9%) [a] | 218 (44.9%) | 0.005 |
| Respiratory symptoms [c] | 65 (92.9%) [a] | 261 (52.2%) | 0.0001 |
| Gastrointestinal symptoms [d] | 38 (54.3%) [a] | 71 (14.2%) | 0.0001 |
| Anosmia | 34 (48.6%)[a] | 42 (8.7%) | 0.0001 |
| Ageusia | 35 (50.0%) [a] | 37 (7.6%) | 0.0001 |
| Rash | 4 (6.1%) | 12 (2.5%) | 0.987 |
| Smoking | 4 (5.7%) [a] | 67 (13.4%) | 0.05 |
| Pneumopathy | 7 (10%) | 43 (8.6%) | 0.698 |
| Diabetes | 3 (4.3%) | 12.0 (2.4%) | 0.214 |
| Hypertension | 5 (7.1%) | 43 (8.6%) | 0.280 |
| Other comorbidities | 2 (2.9%) [a] | 04 (0.8%) | 0.007 |

[a]p<0.05. Continuous variables are presented as mean and standard deviation.

[b]Date of onset of symptoms was not reported by the patients.

[c] Coughing, odynophagia, dyspnea, nasal drainage.

[d] vomiting, diarrhea.

the results from SARS-CoV-2 BAT, compared with the RT-qPCR method as the gold standard, showed 90.2% sensitivity (CI 76.0–97.3%) and 95.5% specificity (CI 90.9–98.2%). The PPV was determined as 84.1% (CI 69.9–93.4%) and the NPV as 97.4% (CI 93.4–99.3%). Again, Fisher's exact test showed a statistically significant association between the results obtained through the two methods (P < 0.0001). The area under the ROC curve calculated only for symptomatic patients who had shown symptoms up to seven days was 0.93 (standard error 0.03; CI 0.87–0.98).

Considering only the asymptomatic patients, the results showed 100.0% sensitivity (CI 39.7–100.0%) and 96.3% specificity (CI 92.1–98.6%), with PPV of 40.0% (CI 12.2–73.8%) and NPV of 100.0% (CI 97.7–100.0%). Fisher's exact test showed a statistically significant association between results (P < 0.0001). The area under the ROC curve calculated only with asymptomatic patients was 0.98 (standard error 0.01; CI 0.96–1.00). The limitation of this method was a false positive rate of 23.5%, indicating that the positive predictive value of the test may be compromised in this sampling.

## Detection of other respiratory viruses

The swab samples collected for RT-qPCR testing of SARS-CoV-2 were also concomitantly tested for a viral panel of 15 other respiratory viruses that have been mostly reported in

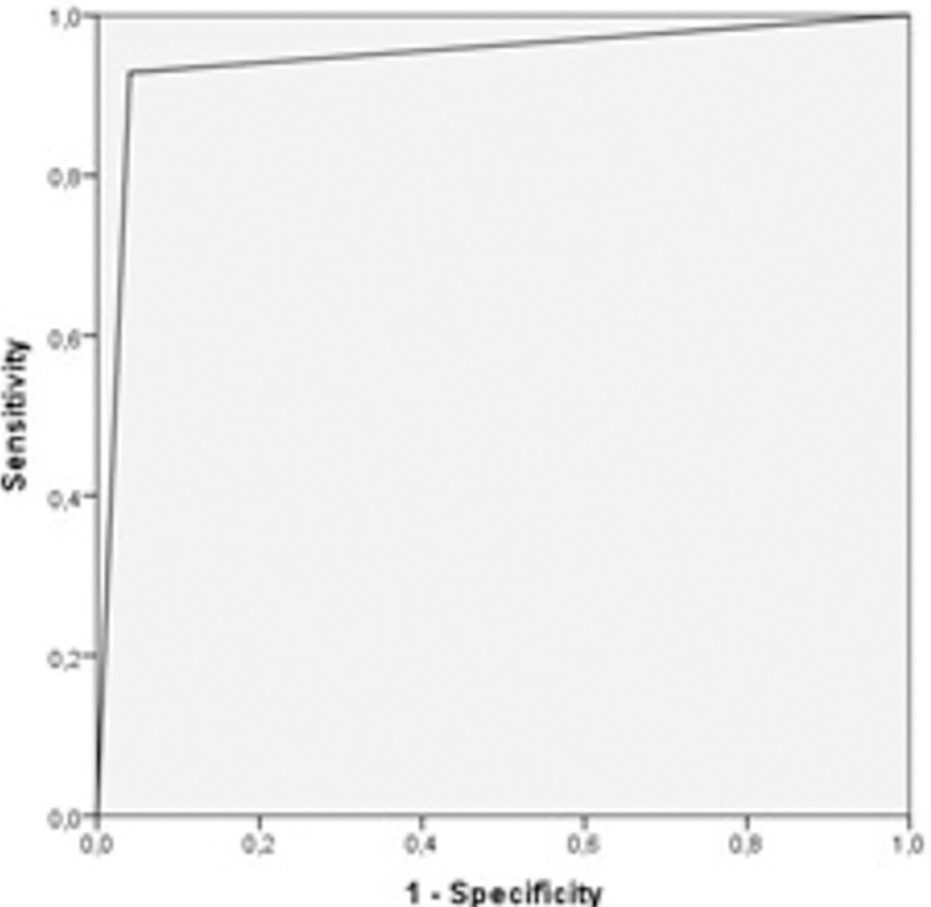

**Fig 3. Full resource spectra for 48 mean positive samples and 48 negative samples.** Light red and light blue colors represent the "distribution" of the sample, while dark colors are the average result. Light gray (not an additional graphic) shows the blend of color shades, representing the standard deviation of the 48 samples mean for positive and negative groups.

Brazilian patients. Among all the patients tested, 32/570 (5.61%) presented coinfection with respiratory viruses identified by RT-qPCR including adenovirus (n = 1; 0.2%); bocavirus (n = 2; 0.4%); other coronaviruses (n = 6; 1.1%); and rhinovirus (n = 23; 4.0%). A single case of coinfection was observed, with simultaneous identification of rhinovirus and another coronavirus. While no symptoms were observed in the 7/32 patients (21.9%) with other respiratory viruses, the symptomatic patients (n = 25; 71.4%) mostly presented respiratory symptoms (n = 22; 88.0%), myalgia and/or arthralgia (n = 14; 56.0%), fever and/or chills (n = 13; 52.0%) and gastrointestinal symptoms (n = 10; 40.0%). No association has been found between false SARS-CoV-2 positives and other respiratory virus infection.

**Table 2. Results from RT-qPCR and BAT for the entire sample.**

|  |  | RT-qPCR SARS-CoV-2 results | | |
|---|---|---|---|---|
|  |  | Positive (%) | Negative (%) | Total (%) |
| Results COVID-19 BAT test | Positive | 65 (11.4) | 20 (3.5) | 85 (14.9) |
|  | Negative | 5 (0.9) | 480 (84.2) | 485 (85.1) |
|  | Total | 70 (12.3) | 500 (87.7) | 570 (100.0) |

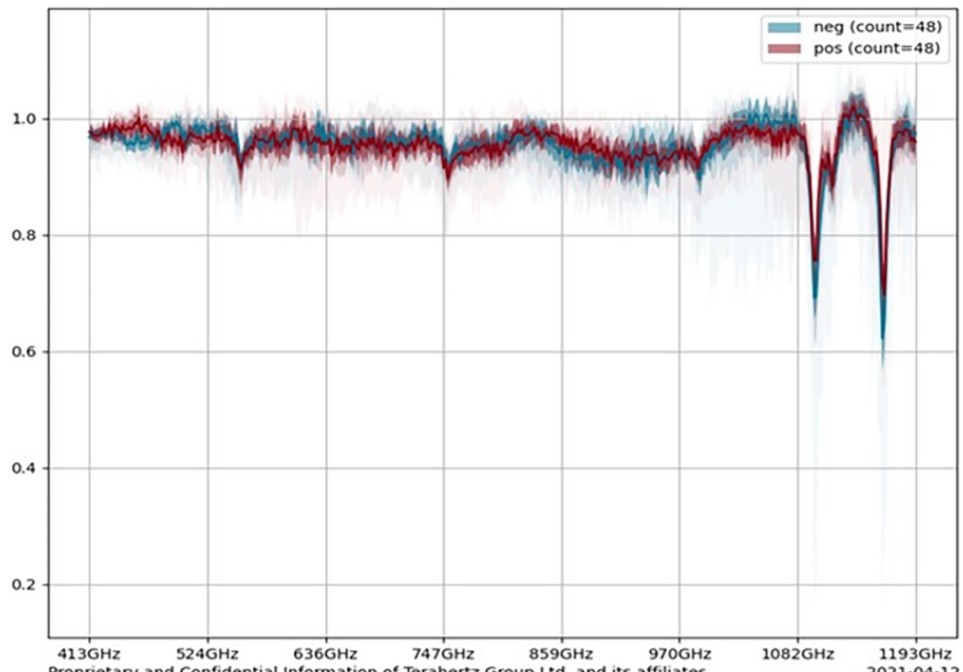

**Fig 4. ROC curve for the entire sample.**

## Discussion

Although RT-qPCR has been very efficient under optimized conditions and considered the gold-standard molecular method for detecting SARS-CoV-2, this method may require sophisticated instruments and qualified personnel [33]. Biological samples may also need to be transported to reference laboratories, thus increasing costs and the time that elapses before processing and delivery of results [34], besides reports of insufficient sampling, false negative and uncertain results [35]. In addition, viable noninvasive sampling alternatives may be desirable to replace oropharynx and nasopharynx swabs, as required for RT-qPCR testing [31]. Although use of saliva has already been reported as alternative form of sampling for virus detection, the diagnosis relies on the patient's stage of infection, with a comparatively narrower window than swabs [27, 34–37] Although several alternative platforms have now been reported, few studies have shown any possible substitute or additional methods that could replace RT-qPCR testing [38, 39].

The results presented here indicated that BAT presented 94.4% accuracy (95% CI: 91 to 98%) for both symptomatic and asymptomatic, which has been in accordance with the recommendations of the WHO [11]. Moreover, its performance has been shown to be superior to that of chest computed tomography used in association with a special algorithm, optimization method for patient treatment [27]. In the present study, the commercial BAT showed to be a fast on-site and easy-to-apply test with superior sensitivity and specificity, compared with serological tests [11, 35, 40] and other methods using saliva [37] and sputum samples [41].

Despite the low frequency of other respiratory viruses, false-positive results were not observed and suggested that coinfection of other respiratory viruses may not compromise the accuracy of the BAT method. Likewise, previous studies have shown no correlation between positive COVID-19 cases and respiratory infections by influenza viruses [42]. As several other

respiratory viruses were absent in the present study, the performance of BAT in such coinfections remain to be fully established.

Rational use of laboratory tools has become crucial for enabling efficient diagnosis, particularly due to the high costs of mass testing [33, 43]. Highly specific, fast, easy-to-apply tests and able to identify most patients at early stages of SARS-CoV-2 infection may promptly ensure patient isolation and other preventive measures against virus transmission and spreading, reducing infection and demand for further testing [11].

Limitations in the present study included the potential bias of symptomatic cases (n = 404; 70.9%), which may have partially compromised the assessment of BAT application for asymptomatic patients who presented positive results. Thus, such parameter should be further evaluated with larger sample number of asymptomatic patients. The 95% CI amplitude for PPV was the widest among all the analytical parameters of performance assessed herein. Further investigations should be also conducted in asymptomatic, underage, and patients in the first days of infection or more than 7 days after clinical onset, with larger sample size to better establish the BAT positive predictive capacity. Finally, the present study has relied on natural occurring cases of SARS-CoV-2 and coinfection with other respiratory viruses at the time of COVID-19 variants and coinfections. Thus, the specific spectral characteristic for each virus and confounding coinfection impact was limited to the income casuistic cases at the time. Thus, future studies should be performed as multicentric surveys, representative samplings and with broader virus coinfection occurrence.

## Conclusion

In summary, comparison between a commercial BAT (TERA.Bio®) and RT-qPCR for diagnosis of SARS-CoV-2 has demonstrated satisfactory accuracy (94.4%) and sensitivity (92.6%) for diagnostic use, except among asymptomatic patients. BAT has also shown 96% specificity for diagnostic use in all patient groups, strongly recommended as screening, fast and noninvasive on-site test, with positive cases confirmed by RT-qPCR. Since the NPV of the BAT method was 98.97%, the commercial BAT may be also indicated as a screening method for ruling out negative cases.

## Supporting information

**S1 Checklist. CONSORT 2010 checklist of information to include when reporting a randomised trial**\*.
(PDF)

**S1 File. Trial study protocol and details of prior approval for human subjects research.**
(PDF)

**S1 Dataset. Data source and results of BAT and TR-PCR to SARS-CoV-2.**
(XLSX)

## Acknowledgments

The authors are grateful to David George Elliff for editing and improving the article.

## Author Contributions

**Conceptualization:** Meila Bastos De Almeida, Regina Aharonov-Nadborny, Eran Gabbai, Leticia Schiavo, Elis Esmanhoto, Jaime Rocha, Ariel Margulis, Marcelo Loureiro.

**Data curation:** Meila Bastos De Almeida, Regina Aharonov-Nadborny, Eran Gabbai, Ana Paula Palka, Leticia Schiavo, Elis Esmanhoto, Irina Riediger, Ariel Margulis, Marcelo Loureiro.

**Formal analysis:** Meila Bastos De Almeida, Regina Aharonov-Nadborny, Eran Gabbai, Ana Paula Palka, Leticia Schiavo, Elis Esmanhoto, Irina Riediger, Marcelo Loureiro.

**Funding acquisition:** Meila Bastos De Almeida, Regina Aharonov-Nadborny, Jaime Rocha.

**Investigation:** Meila Bastos De Almeida, Regina Aharonov-Nadborny, Ana Paula Palka, Leticia Schiavo, Elis Esmanhoto, Irina Riediger, Jaime Rocha, Christina Pettan-Brewer, Louise Bach Kmetiuk, Ivan Roque De Barros-Filho, Alexander Welker Biondo.

**Methodology:** Meila Bastos De Almeida, Regina Aharonov-Nadborny, Leticia Schiavo, Elis Esmanhoto, Irina Riediger, Jaime Rocha, Louise Bach Kmetiuk, Alexander Welker Biondo.

**Project administration:** Meila Bastos De Almeida, Jaime Rocha, Alexander Welker Biondo.

**Resources:** Meila Bastos De Almeida, Christina Pettan-Brewer, Alexander Welker Biondo.

**Software:** Meila Bastos De Almeida, Regina Aharonov-Nadborny.

**Supervision:** Christina Pettan-Brewer, Ivan Roque De Barros-Filho, Alexander Welker Biondo.

**Validation:** Meila Bastos De Almeida, Ana Paula Palka, Leticia Schiavo, Alexander Welker Biondo.

**Visualization:** Meila Bastos De Almeida, Eran Gabbai, Leticia Schiavo.

**Writing – original draft:** Meila Bastos De Almeida, Regina Aharonov-Nadborny, Eran Gabbai, Ana Paula Palka, Leticia Schiavo, Elis Esmanhoto, Irina Riediger, Jaime Rocha, Ariel Margulis, Marcelo Loureiro, Christina Pettan-Brewer, Louise Bach Kmetiuk, Ivan Roque De Barros-Filho, Alexander Welker Biondo.

**Writing – review & editing:** Meila Bastos De Almeida, Regina Aharonov-Nadborny, Eran Gabbai, Ana Paula Palka, Leticia Schiavo, Elis Esmanhoto, Irina Riediger, Jaime Rocha, Ariel Margulis, Marcelo Loureiro, Christina Pettan-Brewer, Louise Bach Kmetiuk, Ivan Roque De Barros-Filho, Alexander Welker Biondo.

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
