## [Decision Letter · Decision Letter 0]

26 Apr 2022

PONE-D-22-07738Clinical trial and detection of SARS-CoV-2 by a commercial breath analysis test based on Terahertz technologyPLOS ONE

Dear Dr. Biondo,

Thank you for submitting your manuscript to PLOS ONE. After careful consideration, we feel that it has merit but does not fully meet PLOS ONE’s publication criteria as it currently stands. Therefore, we invite you to submit a revised version of the manuscript that addresses the points raised during the review process.

ACADEMIC EDITOR:There are some minor comments presented by reviewers. Please revise according to the suggestions of the reviewers or write a detailed rebuttal on a point-by-point basis.

We look forward to receiving your revised manuscript.

Kind regards,

Davor Plavec, MD, MSc, PhD, Prof.

Academic Editor

PLOS ONE

Journal Requirements:

Additional Editor Comments:

Please revise according to the suggestions of the reviewers or write a detailed rebuttal on a point-by-point basis.

Reviewers' comments:

Reviewer's Responses to Questions

**Comments to the Author**

1. Is the manuscript technically sound, and do the data support the conclusions?

Reviewer #1: Yes

Reviewer #2: Yes

Reviewer #3: Yes

2. Has the statistical analysis been performed appropriately and rigorously? 

Reviewer #1: I Don't Know

Reviewer #2: Yes

Reviewer #3: Yes

3. Have the authors made all data underlying the findings in their manuscript fully available?

Reviewer #1: Yes

Reviewer #2: Yes

Reviewer #3: Yes

4. Is the manuscript presented in an intelligible fashion and written in standard English?

Reviewer #1: Yes

Reviewer #2: Yes

Reviewer #3: Yes

5. Review Comments to the Author

Reviewer #1: Interesting results that should be tested in the broader population and different circumstances but this work is pointing in the direction that this system can be used in Covid 19 pandemic and maybe for the further possible epidemics. The test for discrimination between the respiratory infection and specific respiratory infection is warranted.

Reviewer #2: In the paper authors presented results of clinical trial in which evaluate a commercial breath analysis test (TERA.Bio®) for detecting the SARS-CoV-2 present in exhaled air samples of suspicious persons.

All components of the manuscript are correctly presented, including introduction, results, statistical analysis, discussion and conclusions.

However, the study conducted has certain limitations;

1. the potential bias of symptomatic cases which may have partially compromised the assessment of breath analysis thest (BAT) for asymptomatic patients

2. the BAT did not tested in asymptomatic patients and patients in the first days of infection or after 7 days of the onset of symptoms

3. the small sample size of infected persons (70 confirmed infected persons)

4.relatively limited detection of occurrence of other respiratory viruses, or other pathogens (including bacterial), for evidence of co-infection in patinets with COVID-19 .

All these limitations were anticipated by the authors and suggested how to improve them.

Reviewer #3: This is a well-written paper with innovations in SARS-COV2 diagnostics. In the Materials and Methods section, it would be good to explain the detailed procedure as it relates to the collection of exhaled air samples to prevent the spread of SARS-CoV2. It is visible that it is done in disposable air collection kit, but can it be sampled next to the patient (since these are all outpatient samples, not from hospialized patients). What about safety to the enviroment during the sampling? Does the time between sampling and analysis influences the results?

6. PLOS authors have the option to publish the peer review history of their article (what does this mean?). If published, this will include your full peer review and any attached files.

Reviewer #1: No

Reviewer #2: No

Reviewer #3: No

---

## [Author Response · Author response to Decision Letter 0]

3 Jun 2022

Manuscript #PONE-D-22-07738

Title “Clinical trial and detection of SARS-CoV-2 by a commercial breath analysis test based on Terahertz technology”

1. Thank you for stating the following Funding Information in your manuscript:

"Brazilian National Council for Scientific and Technological Development -CNPq (material support) (2020-1/402341- AWB and CPB). The funders had no role in study design, data collection and analysis, decision to publish, or preparation of the manuscript."

We note that you have provided funding information that is currently declared in your Fund-ing Statement. However, funding information should not appear in any areas of your manu-script. We will only publish funding information present in the Funding Statement section of the online submission form.

Please remove any funding-related text from the manuscript.

Answer: Funding information was removed from the manuscript.

Review Comments to the Author

Please use the space provided to explain your answers to the questions above. You may also include additional comments for the author, including concerns about dual publication, research ethics, or publication ethics. (Please upload your review as an attachment if it ex-ceeds 20,000 characters)

Reviewer #1: Interesting results that should be tested in the broader population and differ-ent circumstances, but this work is pointing in the direction that this system can be used in Covid 19 pandemic and maybe for the further possible epidemics. The test for discrimina-tion between the respiratory infection and specific respiratory infection is warranted.

Answer: Authors are thankful to reviewer for the positive comment.

Reviewer #2: In the paper authors presented results of clinical trial in which evaluate a commercial breath analysis test (TERA.Bio®) for detecting the SARS-CoV-2 present in exhaled air samples of suspicious persons. All components of the manuscript are correctly presented, including introduction, results, statistical analysis, discussion and conclusions.

Answer: Authors are thankful to reviewer for the positive comment.

However, the study conducted has certain limitations;

1. the potential bias of symptomatic cases which may have partially compromised the as-sessment of breath analysis thest (BAT) for asymptomatic patients

2. the BAT did not tested in asymptomatic patients and patients in the first days of infection or after 7 days of the onset of symptoms

3. the small sample size of infected persons (70 confirmed infected persons)

4.relatively limited detection of occurrence of other respiratory viruses, or other pathogens (including bacterial), for evidence of co-infection in patients with COVID-19 .

All these limitations were anticipated by the authors and suggested how to improve them.

Answer: Authors are thankful to reviewer for the positive comment, since all above limita-tions have already been reported and discussed intext.

Reviewer #3 (green intext): This is a well-written paper with innovations in SARS-COV2 diagnostics. In the Materials and Methods section, it would be good to explain the detailed procedure as it relates to the collection of exhaled air samples to prevent the spread of SARS-CoV2. 

Changed: Reviewer is right, and information was added.

Now you read: “The study protocol has included non-hospitalized patients who were re-ferred by the Curitiba city health professionals after suspicious RT-qPCR results. Thus, exhaled air samples and nasopharyngeal swab samples were concomitantly collected and tested by the official city service. Each patient sampled by nasopharyngeal swab was also given a dischargeable kit of exhaled air analyzer containing an identified sampling device, which was uncovered in both sides with opened capsule and the melamine membrane in-ternally coupled. Patients were then asked to keep a 2-meter distance from the operator, swallow all saliva in the mouth and deeply blow the device for 5 times. After blown, the pa-tient was asked to close both device sides with the correspondent lids (within the kit) until hear a click noise of sealed lid, capturing the exhaled breath constituents inside the device. The operator received the device, inspected the complete lid closure, and disinfected the external parts with 70% alcohol. After taken to the laboratory, device was opened and cap-sule with internal membrane removed and inserted into the equipment for analysis. Sam-ples were collected up to 6 hours maximum after sampling, which insured best readings due to membrane integrity, ideal absorption of electromagnetic wave and wave propagation into the exhaled air sample. After analysis, device was discharged as hospital contaminated garbage.” (Page 9-10, Lines 238-258).

It is visible that it is done in disposable air collection kit, but can it be sampled next to the patient (since these are all outpatient samples, not from hospitalized patients). What about safety to the environment during the sampling? Does the time between sampling and anal-ysis influences the results?

Changed: Reviewer is right, and information was added.

Now you read: “The study protocol has included non-hospitalized patients who were re-ferred by the Curitiba city health professionals after suspicious RT-qPCR results. Thus, exhaled air samples and nasopharyngeal swab samples were concomitantly collected and tested by the official city service. Each patient sampled by nasopharyngeal swab was also given a dischargeable kit of exhaled air analyzer containing an identified sampling device, which was uncovered in both sides with opened capsule and the melamine membrane in-ternally coupled. Patients were then asked to keep a 2-meter distance from the operator, swallow all saliva in the mouth and deeply blow the device for 5 times. After blown, the pa-tient was asked to close both device sides with the correspondent lids (within the kit) until hear a click noise of sealed lid, capturing the exhaled breath constituents inside the device. The operator received the device, inspected the complete lid closure, and disinfected the external parts with 70% alcohol. After taken to the laboratory, device was opened and cap-sule with internal membrane removed and inserted into the equipment for analysis. Sam-ples were collected up to 6 hours maximum after sampling, which insured best readings due to membrane integrity, ideal absorption of electromagnetic wave and wave propagation into the exhaled air sample. After analysis, device was discharged as hospital contaminated garbage.” (Page 9-10, Lines 238-258).

---

## [Decision Letter · Decision Letter 1]

16 Jun 2022

PONE-D-22-07738R1Clinical trial and detection of SARS-CoV-2 by a commercial breath analysis test based on Terahertz technologyPLOS ONE

Dear Dr. Biondo,

Thank you for submitting your manuscript to PLOS ONE. After careful consideration, we feel that it has merit but does not fully meet PLOS ONE’s publication criteria as it currently stands. Therefore, we invite you to submit a revised version of the manuscript that addresses the points raised during the review process.

Please revise as suggested by the reviewer.

We look forward to receiving your revised manuscript.

Kind regards,

Davor Plavec, MD, MSc, PhD, Prof.

Academic Editor

PLOS ONE

Journal Requirements:

Additional Editor Comments:

Please revise as suggested by the reviewer.

Reviewers' comments:

Reviewer's Responses to Questions

**Comments to the Author**

1. If the authors have adequately addressed your comments raised in a previous round of review and you feel that this manuscript is now acceptable for publication, you may indicate that here to bypass the “Comments to the Author” section, enter your conflict of interest statement in the “Confidential to Editor” section, and submit your "Accept" recommendation.

Reviewer #4: (No Response)

2. Is the manuscript technically sound, and do the data support the conclusions?

Reviewer #4: Yes

3. Has the statistical analysis been performed appropriately and rigorously? 

Reviewer #4: Yes

4. Have the authors made all data underlying the findings in their manuscript fully available?

Reviewer #4: Yes

5. Is the manuscript presented in an intelligible fashion and written in standard English?

Reviewer #4: Yes

6. Review Comments to the Author

Reviewer #4: A clinical trial was conducted which aimed to evaluate the commercial breath analysis test TERA.Bio (r) and a deterministic algorithm for detecting SARS-CoV-2 compounds exhaled in air samples. Results were compared to RT-qPCR as the gold standard. The sensitivity and specificity of the test was 92.6% and 96.0%, respectively.

Minor revisions:

1- State and justify the study’s target sample size with a pre-study statistical power calculation.

2- In the statistical analysis section indicate the methods used to compare the features listed in table 1. Provide more precise p-values rather than p < 0.05.

3- Lines 364-370 and subsequent sections: In the statistical analysis section, state the method used to calculate the confidence intervals.

4- Table 2: In addition to frequencies, provide the corresponding percentages.

7. PLOS authors have the option to publish the peer review history of their article (what does this mean?). If published, this will include your full peer review and any attached files.

Reviewer #4: No

---

## [Author Response · Author response to Decision Letter 1]

1 Jul 2022

Re: PLOS ONE Decision: Revision required [PONE-D-22-07738R1]

Dear Dr Plavec,

Please find attached our reviewed manuscript entitled “Clinical trial and detection of SARS-CoV-2 by a commercial breath analysis test based on Terahertz technology” [PONE-D-22-07738R1]. All the corrections and suggestions indicated by the reviewers were fully ad-dressed, accepted, and rewritten.

A response letter has been added to pinpoint the editor and reviewers’ requests, with a detailed list of corrections and a description of changes made within the manuscript. To better identify the corrections of each reviewer, changes were highlighted in different colors in the revised manuscript. Thank you for the opportunity and please do not hesitate to con-tact us with any further question.

Sincerely,

Alexander Welker Biondo, DMV, MSc, PhD.

Professor, Department of Veterinary Medicine, UFPR, Brazil

Visiting Professor, Purdue University, IN, USA

E-mail: abiondo@ufpr.br / Phone: +55 (41) 3350-5812

Manuscript #PONE-D-22-07738R1

Title “Clinical trial and detection of SARS-CoV-2 by a commercial breath analysis test based on Terahertz technology”

Reviewer #4: A clinical trial was conducted which aimed to evaluate the commercial breath analysis test TERA.Bio (r) and a deterministic algorithm for detecting SARS-CoV-2 com-pounds exhaled in air samples. Results were compared to RT-qPCR as the gold standard. The sensitivity and specificity of the test was 92.6% and 96.0%, respectively.

Minor revisions:

1- State and justify the study’s target sample size with a pre-study statistical power calcula-tion.

Changed: Reviewer is right, that information was missing. Explanation was included in the Study Sample Size and Data Validation Section of the manuscript. The correspondent ref-erence was cited in the sentence.

Now you read: “The assumed positivity of RT-qPCR for sampling calculation in the clinical trial herein was 25%, based on epidemiological COVID-19 reports of Curitiba city, which had an estimated population of approximately 1.9 million habitants at the time. Aiming at a type I error (α) of 0.05, and an estimated precision of 0.05, a minimum sample size of 400 subjects was obtained to statistically evaluate the BAT test, considering a 90% sensitivity as minimum for a useful diagnostic test (30). Representing a consecutive, casuistic, ran-dom, and homogeneous sampling, collected from all subjects who presented themselves at the hospital in the period, a total of 570 successful samplings were included in the present study.” (Page 7-8, Lines 173-183).

2- In the statistical analysis section indicate the methods used to compare the features listed in table 1. Provide more precise p-values rather than p < 0.05.

Changed: The sentences were included in the Statistical Analysis section of the manu-script.

P values were included, whenever higher than 0.0001. P values lower than 0.0001 are not expressed by the software used for statistical analysis (SPSS).

Now you read: “Sociodemographic, epidemiological, and clinical characteristics were pre-sented as percentages, alongside their 95% proportion confidence intervals, and by arith-metic mean and standard deviation. The results were presented in contingency tables, al-lowing the calculation of sensitivity, specificity, and predictive values for the BAT. Assess-ment of the association between categorical variables; and the BATs and RT-qPCR tests was performed by Fisher’s exact test. Continuous variables were compared using Student’s unpaired t-test. The precision of the BAT was assessed using ROC curve analysis. Statisti-cal significance was assumed for P<0.05.

In addition, machine learning was based on “training data” that had been collected prior to the study herein. Over 4,000 subjects were tested using the BATs at several loca-tions worldwide including Brazil. The THz spectrum features for healthy and infected (nega-tive and positive) subjects were processed using ML techniques, to establish a mathemati-cal algorithm for COVID-19 classification. The statistical analyses were performed using the SPSS version 17 software.” (Page 11-12, Lines 290-306).

3- Lines 364-370 and subsequent sections: In the statistical analysis section, state the method used to calculate the confidence intervals.

Changed: The information was included in the Statistical analysis Section of the manu-script.

Now you read: “The results from the RT-qPCR test and SARS-CoV-2 BAT for all subjects was presented (Table 2). Using RT-qPCR method as the gold standard, the commercial BAT method was found to have 92.7% sensitivity (CI 84.1-97.6%) and 96.0% specificity (CI 93.9-97.5%). The positive predictive value (PPV) and negative predictive value (NPV) were determined as 76.5% (CI 66.3-85.0%) and 99.0% (CI 97.6-99.7%), respectively. Fisher's exact test showed a statistically significant association between results (P < 0.0001). The area under the ROC curve of the total sampling was 0.94 (SD 0.19; CI 0.91-0.98) (Figure 4).

Figure 4. ROC curve for the entire sample.

Considering only the symptomatic patients and using RT-qPCR as the gold standard, BAT presented 92.4% sensitivity (CI 83.2-97.5%) and 95.9% specificity (CI 93.1-97.7%). The PPV was determined as 81.3% (CI 70.7-89.4%) and the NPV as 98.5% (CI 96.5-99.5%). Fisher's exact test showed a statistically significant association between results (p<0.0001). The area under the ROC curve calculated only with symptomatic patients was 0.94 (standard error 0.20; CI 0.90-0.98). 

Given that the optimum sampling window of opportunity for RT-qPCR has been reported to comprise the first seven days with symptoms, the data were stratified for analysis including the symptomatic patients who had shown symptoms for up to seven days. In such scenario, the results from SARS-CoV-2 BAT, compared with the RT-qPCR method as the gold standard, showed 90.2% sensitivity (CI 76.0-97.3%) and 95.5% specificity (CI 90.9-98.2%). The PPV was determined as 84.1% (CI 69.9-93.4%) and the NPV as 97.4% (CI 93.4-99.3%). Again, Fisher's exact test showed a statistically significant association between the results obtained through the two methods (P < 0.0001). The area under the ROC curve calculated only for symptomatic patients who had shown symptoms up to seven days was 0.93 (standard error 0.03; CI 0.87-0.98). 

Considering only the asymptomatic patients, the results showed 100.0% sensitivity (CI 39.7-100.0%) and 96.3% specificity (CI 92.1- 98.6%), with PPV of 40.0% (CI 12.2-73.8%) and NPV of 100.0% (CI 97.7-100.0%). Fisher's exact test showed a statistically significant association between results (P < 0.0001). The area under the ROC curve calculated only with asymptomatic patients was 0.98 (standard error 0.01; CI 0.96-1.00). The limitation of this method was a false positive rate of 23.5%, indicating that the positive predictive value of the test may be compromised in this sampling.” (Page 15-17, Lines 358-401).

4- Table 2: In addition to frequencies, provide the corresponding percentages.

Changed: The percentages were included in Table 2.

---

## [Decision Letter · Decision Letter 2]

11 Jul 2022

PONE-D-22-07738R2Clinical trial and detection of SARS-CoV-2 by a commercial breath analysis test based on Terahertz technologyPLOS ONE

Dear Dr. Biondo,

Thank you for submitting your manuscript to PLOS ONE. After careful consideration, we feel that it has merit but does not fully meet PLOS ONE’s publication criteria as it currently stands. Therefore, we invite you to submit a revised version of the manuscript that addresses the points raised during the review process.

ACADEMIC EDITOR:

Please make suggested corrections.

We look forward to receiving your revised manuscript.

Kind regards,

Davor Plavec, MD, MSc, PhD, Prof.

Academic Editor

PLOS ONE

Journal Requirements:

Additional Editor Comments :

Dear Authors, please make corrections suggested by the reviewer.

Reviewers' comments:

Reviewer's Responses to Questions

**Comments to the Author**

1. If the authors have adequately addressed your comments raised in a previous round of review and you feel that this manuscript is now acceptable for publication, you may indicate that here to bypass the “Comments to the Author” section, enter your conflict of interest statement in the “Confidential to Editor” section, and submit your "Accept" recommendation.

Reviewer #4: (No Response)

2. Is the manuscript technically sound, and do the data support the conclusions?

Reviewer #4: Yes

3. Has the statistical analysis been performed appropriately and rigorously? 

Reviewer #4: Yes

4. Have the authors made all data underlying the findings in their manuscript fully available?

Reviewer #4: Yes

5. Is the manuscript presented in an intelligible fashion and written in standard English?

Reviewer #4: Yes

6. Review Comments to the Author

Reviewer #4: Minor revisions: (Note that line numbers refer to those in the track changes version of the manuscript.)

1- Line 183: In the sample size justification section, state the statistical power that was attained.

2- Table 1: Some of the p-values display a comma in place of a decimal. Perhaps these errors resulted when the document was formatted by the journal.

3- In the "Statistical analysis" section, list and describe the statistical methods used to estimate the p-values shown in Table 1. Also indicate the statistical methods used to generate 95% confidence intervals.

7. PLOS authors have the option to publish the peer review history of their article (what does this mean?). If published, this will include your full peer review and any attached files.

Reviewer #4: No

---

## [Author Response · Author response to Decision Letter 2]

16 Jul 2022

1- Line 183: In the sample size justification section, state the statistical power that was attained.

Changed: The statistical power was included.

Now you read: “Aiming at a type I error (α) of 0.05, and an estimated precision of 0.05, a minimum sample size of 400 subjects was obtained to statistically evaluate the BAT test, considering a 90% sensitivity as minimum for a useful diagnostic test (30). Representing a consecutive, casuistic, random, and homogeneous sampling, collected from all subjects who presented themselves at the hospital in the period, a total of 570 successful samplings were included in the present study, with statistical power of 0.819.” (Page 8, Lines 193).

2- Table 1: Some of the p-values display a comma in place of a deci-mal. Perhaps these errors resulted when the document was formatted by the journal.

Changed: Number have been corrected at table 1.

3- In the "Statistical analysis" section, list and describe the statistical methods used to estimate the p-values shown in Table 1. Also indicate the statistical methods used to generate 95% confidence intervals.

Changed: Statistical methods were inserted.

Now you read: “Student's t-test was used to determine estimate the p-values and chi-square test to generate 95% confidence intervals.” (Page 11, Lines 299-301).

---

## [Decision Letter · Decision Letter 3]

20 Jul 2022

PONE-D-22-07738R3Clinical trial and detection of SARS-CoV-2 by a commercial breath analysis test based on Terahertz technologyPLOS ONE

Dear Dr. Biondo,

Thank you for submitting your manuscript to PLOS ONE. After careful consideration, we feel that it has merit but does not fully meet PLOS ONE’s publication criteria as it currently stands. Therefore, we invite you to submit a revised version of the manuscript that addresses the points raised during the review process.

ACADEMIC EDITOR: Dear Authors, please consult with the statistician to improve the text of your Statistical analysis as this is obviously the problem for you and the root of corrections. 

We look forward to receiving your revised manuscript.

Kind regards,

Davor Plavec, MD, MSc, PhD, Prof.

Academic Editor

PLOS ONE

Journal Requirements:

Additional Editor Comments:

Dear Authors, please consult with the statistician to improve the text of your Statistical analysis as this is obviously the problem for you and the root of corrections.

Reviewers' comments:

Reviewer's Responses to Questions

**Comments to the Author**

1. If the authors have adequately addressed your comments raised in a previous round of review and you feel that this manuscript is now acceptable for publication, you may indicate that here to bypass the “Comments to the Author” section, enter your conflict of interest statement in the “Confidential to Editor” section, and submit your "Accept" recommendation.

Reviewer #4: (No Response)

2. Is the manuscript technically sound, and do the data support the conclusions?

Reviewer #4: Yes

3. Has the statistical analysis been performed appropriately and rigorously? 

Reviewer #4: Yes

4. Have the authors made all data underlying the findings in their manuscript fully available?

Reviewer #4: Yes

5. Is the manuscript presented in an intelligible fashion and written in standard English?

Reviewer #4: Yes

6. Review Comments to the Author

Reviewer #4: Minor Revisions:

The recently added sentence at line 299 is not clear: "Student's t-test was used to determine estimate the p-values and chi-square test to generate 95% confidence intervals." A student's t-test is used for comparing the mean differences in two groups. A chi-square tests is used for comparing differences in proportions. Table 1 summarizes both means and proportions. Furthermore, the student's unpaired t-test is mentioned earlier in this paragraph. Improve the clarity of this first paragraph in the "Statistical analysis" section.

7. PLOS authors have the option to publish the peer review history of their article (what does this mean?). If published, this will include your full peer review and any attached files.

Reviewer #4: No

---

## [Author Response · Author response to Decision Letter 3]

24 Jul 2022

Re: PLOS ONE Decision: Revision required [PONE-D-22-07738R3]

Dear Dr Plavec,

Please find attached our reviewed manuscript entitled “Clinical trial and detection of SARS-CoV-2 by a commercial breath analysis test based on Terahertz technology” [PONE-D-22-07738R3]. All the corrections and suggestions indicated by the reviewers were fully ad-dressed, accepted, and rewritten.

A response letter has been added to pinpoint the editor and reviewers’ requests, with a detailed list of corrections and a description of changes made within the manuscript. To better identify the corrections of each reviewer, changes were highlighted in different colors in the revised manuscript. Thank you for the opportunity and please do not hesitate to con-tact us with any further question.

Manuscript #PONE-D-22-07738R3

Title “Clinical trial and detection of SARS-CoV-2 by a commercial breath analysis test based on Terahertz technology”

Reviewer #4: Minor Revisions:

The recently added sentence at line 299 is not clear: "Student's t-test was used to determine estimate the p-values and chi-square test to generate 95% confidence intervals." A student's t-test is used for com-paring the mean differences in two groups. A chi-square tests is used for comparing differences in proportions. Table 1 summarizes both means and proportions. Furthermore, the student's unpaired t-test is mentioned earlier in this paragraph. Improve the clarity of this first par-agraph in the "Statistical analysis" section.

Changed: Sentence was rewritten to clarify.

Now you read: “Sociodemographic, epidemiological, and clinical char-acteristics were presented as percentages, arithmetic means and standard deviations. Differences in proportions were compared by chi-square test with calculated 95% confidence intervals. The results were presented in contingency tables, allowing the calculation of sensitivity, specificity, and predictive values for the BAT. Assessment of associa-tion between categorical variables, and BAT and RT-qPCR tests was performed by Fisher’s exact test. Mean differences in variables were compared by p-values of Student’s unpaired t-test, assuming statistical significance when p<0.05. Precision of the BAT was assessed using ROC curve analysis.” (Page 11, Lines 290-300).

---

## [Decision Letter · Decision Letter 4]

10 Aug 2022

Clinical trial and detection of SARS-CoV-2 by a commercial breath analysis test based on Terahertz technology

PONE-D-22-07738R4

Dear Dr. Biondo,

We’re pleased to inform you that your manuscript has been judged scientifically suitable for publication and will be formally accepted for publication once it meets all outstanding technical requirements.

Kind regards,

Davor Plavec, MD, MSc, PhD, Prof.

Academic Editor

PLOS ONE

Additional Editor Comments (optional):

Reviewers' comments:

Reviewer's Responses to Questions

**Comments to the Author**

1. If the authors have adequately addressed your comments raised in a previous round of review and you feel that this manuscript is now acceptable for publication, you may indicate that here to bypass the “Comments to the Author” section, enter your conflict of interest statement in the “Confidential to Editor” section, and submit your "Accept" recommendation.

Reviewer #4: All comments have been addressed

2. Is the manuscript technically sound, and do the data support the conclusions?

Reviewer #4: (No Response)

3. Has the statistical analysis been performed appropriately and rigorously? 

Reviewer #4: (No Response)

4. Have the authors made all data underlying the findings in their manuscript fully available?

Reviewer #4: (No Response)

5. Is the manuscript presented in an intelligible fashion and written in standard English?

Reviewer #4: (No Response)

6. Review Comments to the Author

Reviewer #4: (No Response)

7. PLOS authors have the option to publish the peer review history of their article (what does this mean?). If published, this will include your full peer review and any attached files.

Reviewer #4: No

---

## [Editor Report · Acceptance letter]

12 Sep 2022

PONE-D-22-07738R4 

Clinical trial and detection of SARS-CoV-2 by a commercial breath analysis test based on Terahertz technology 

Dear Dr. Biondo:

I'm pleased to inform you that your manuscript has been deemed suitable for publication in PLOS ONE. Congratulations! Your manuscript is now with our production department. 

Kind regards, 

on behalf of

Dr. Davor Plavec 

Academic Editor

PLOS ONE